# Impact of Advanced Paternal Age on Fertility and Risks of Genetic Disorders in Offspring

**DOI:** 10.3390/genes14020486

**Published:** 2023-02-14

**Authors:** Aris Kaltsas, Efthalia Moustakli, Athanasios Zikopoulos, Ioannis Georgiou, Fotios Dimitriadis, Evangelos N. Symeonidis, Eleftheria Markou, Theologos M. Michaelidis, Dung Mai Ba Tien, Ioannis Giannakis, Eleni Maria Ioannidou, Athanasios Papatsoris, Panagiota Tsounapi, Atsushi Takenaka, Nikolaos Sofikitis, Athanasios Zachariou

**Affiliations:** 1Laboratory of Spermatology, Department of Urology, Faculty of Medicine, School of Health Sciences, University of Ioannina, 45110 Ioannina, Greece; 2Laboratory of Medical Genetics in Clinical Practice, Faculty of Medicine, School of Health Sciences, University of Ioannina, 45110 Ioannina, Greece; 3Department of Urology, Faculty of Medicine, School of Health Sciences, Aristotle University of Thessaloniki, 54124 Thessaloniki, Greece; 4Department of Microbiology, University Hospital of Ioannina, 45500 Ioannina, Greece; 5Department of Biological Applications and Technologies, School of Health Sciences, University of Ioannina, 45110 Ioannina, Greece; 6Biomedical Research Institute, Foundation for Research and Technology-Hellas, 45500 Ioannina, Greece; 7Department of Andrology, Binh Dan Hospital, Ho chi Minh City 70000, Vietnam; 8Dental Medicine, Medical University of Sofia, 1431 Sofia, Bulgaria; 92nd Department of Urology, School of Medicine, Sismanoglio Hospital, National and Kapodistrian Univesity of Athens, 15126 Athens, Greece; 10Division of Urology, Department of Surgery, School of Medicine, Faculty of Medicine, Tottori University, Yonago 683-8503, Japan

**Keywords:** genetics, paternal age, infertility, offspring, assisted reproductive techniques

## Abstract

The average age of fathers at first pregnancy has risen significantly over the last decade owing to various variables, including a longer life expectancy, more access to contraception, later marriage, and other factors. As has been proven in several studies, women over 35 years of age have an increased risk of infertility, pregnancy problems, spontaneous abortion, congenital malformations, and postnatal issues. There are varying opinions on whether a father’s age affects the quality of his sperm or his ability to father a child. First, there is no single accepted definition of old age in a father. Second, much research has reported contradictory findings in the literature, particularly concerning the most frequently examined criteria. Increasing evidence suggests that the father’s age contributes to his offspring’s higher vulnerability to inheritable diseases. Our comprehensive literature evaluation shows a direct correlation between paternal age and decreased sperm quality and testicular function. Genetic abnormalities, such as DNA mutations and chromosomal aneuploidies, and epigenetic modifications, such as the silencing of essential genes, have all been linked to the father’s advancing years. Paternal age has been shown to affect reproductive and fertility outcomes, such as the success rate of in vitro fertilisation (IVF), intracytoplasmic sperm injection (ICSI), and premature birth rate. Several diseases, including autism, schizophrenia, bipolar disorders, and paediatric leukaemia, have been linked to the father’s advanced years. Therefore, informing infertile couples of the alarming correlations between older fathers and a rise in their offspring’s diseases is crucial, so that they can be effectively guided through their reproductive years.

## 1. Introduction

A Longer life expectancy, later marriage ages, greater median wages, and a change in the social standing of women are all factors that cause couples to delay having children [1]. Assisted reproductive technologies have increased the average paternal age at first delivery by allowing older parents with a lower chance of having a safe pregnancy to conceive [2]. Several factors are affected by paternal age, including testicular function [3], reproductive hormones [4], sperm parameters [5], sperm DNA integrity [6,7], telomere length [8], de novo mutation rate [9], chromosome structure [6], and epigenetic factors [10].

There is a rise in anomalies during pregnancy and even foetal mortality due to these changes, which harm fertility and reproductive outcomes in middle-aged and elderly couples [11]. In addition, advanced paternal age is associated with a wide range of health problems in the offspring, including skeletal dysplasia (achondroplasia, thanatophoric dysplasia, osteogenesis imperfecta) [12], psychiatric morbidity (autism, attention-deficit/hyperactivity disorder, psychosis, bipolar disorder, suicide attempt, and substance use problem) [13] and academic morbidity (failing grades and low educational attainment) [13] (Figure 1). This review will focus on the clinical outcomes associated with the molecular impacts of paternal age. The risk of postponing fatherhood and how it can impact an older couple’s ability to have children and their general quality of life are issues that we wish to bring to the attention of physicians and older couples.

## 2. Advanced Paternal Age Effects on Testicular Function and Sperm Quality

### 2.1. Reproduction and Male Hormones: The Testes’ Crucial Role

Testicular function decreases with ageing, according to many studies conducted in recent years [14,15,16,17]. However, the molecular and genetic basis of testicular ageing has not been adequately elucidated. Using single-cell RNA-seq, Nie et al. analysed age-related changes in germline development and somatic cells of the testis in more than 44,000 cells from young and older men [18]. There was evidence of age-related increases in modest changes to spermatogonial stem cells but moderate to severe dysregulations of spermatogenesis and somatic cells. Of note, the extent of the dysregulation was related to the body mass index in older men but not in younger men [18]. According to research by Handelsman et al., males 80 years old and older have increasingly smaller testicles as their age increases [3]. Based on their findings, Mahmoud et al. concluded that males over 75 years of age have a mean testicular volume 31% less than men aged 18–40 years [19]. If the Sertoli cell count drops, the testicular volume will follow suit [19]. During spermatogenesis, Sertoli cells, which perform endocrine and paracrine functions, provide the germ cells with nutrients and structural support [20]. According to Johnson and colleagues, the basal membrane of the seminiferous tubules similarly thickens with age [15]. Senile testicular abnormalities such as hernia-like protrusions, reduced spermatogenesis, and a thicker basement membrane have all been linked to insufficient blood flow [21].

The most frequent clinically relevant changes linked to male ageing include rising follicle-stimulating hormone (FSH) serum levels and decreased testosterone levels [22]. Furthermore, male andropause symptoms such as reduced libido, exhaustion and memory loss have been related to declining testosterone levels [23]. In addition, declining libido and sexual activity are associated with ageing, which may make many men infertile [24,25].

To produce testosterone, Leydig cells are required. Testicles of older fathers tend to have fewer Leydig cells [14]. Neaves et al. found that compared to the number found in men in the age range from 20 to 48 years, the average number of Leydig cell nuclei drops by 50% in men between 50 and 76 years [14]. A significant factor in the onset of andropause and its accompanying disease in older men is the gradual depletion of Leydig cells [26]. A man’s total testosterone and free testosterone (1.2%) levels start to fall after the age of 50 and beyond, which is at least partly attributable to a drop in Leydig cells [27].

According to Wu et al., alterations in the hypothalamic–pituitary–testicular (HPT) axis disrupt the average production and control of many reproductive hormones, leading to atrophy of the testes as people age [28]. In addition, there is proof that alterations in many hormone levels are related to a father’s older age.

### 2.2. Parameters of Semen Analysis

Impaired spermatogenesis, abnormal sperm, sperm dysfunction, and Sertoli and Leydig cell impairment are more common in the testes of older men, which may ultimately lead to male infertility [29]. A patient’s sperm must be tested as a crucial initial step in diagnosing male infertility. According to recommendations, tests should be performed to measure ejaculate volume as well as sperm concentration, motility, and morphology [30]. There is a clear drop in paternal age-related sperm volume, sperm motility, and normal morphology percentage [5,31]. The American Urological Association advises starting the diagnostic process by carefully reviewing the patient’s reproductive history and the outcomes of two meticulously performed semen tests [32]. The analysis of a sperm sample may reveal the necessity for further investigation. Indeed, the European Association of Urology (EAU) recommends this if semen abnormalities are detected [33]. Age-related alterations in a man’s sperm have been linked in several ways [34]. These alterations may be related to seminal vesicle insufficiency, which reduces the amount of semen produced, or to prostate atrophy, which lowers the amount of water and protein in the ejaculate, reducing the amount of ejaculate and the mobility of sperm [34]. Kidd et al. discovered that although sperm concentration did not change with age, sperm morphology and motility did [34]. Between the ages of 30 to 50 years, significant differences in sperm volume (3–22%), motility (3–37%), and morphology (4–18%) have been observed [34]. Hossain and colleagues discovered that the volume and count decline with fathers’ ageing [35]. There is evidence that older men produce sperm that is less motile in a lower volume [36]. Age-related variations were detected in a large prospective study of 3729 male partners tested for semen quality [36]. Over 5000 sperm samples from males aged 16 to 72 years were analysed to ascertain how male ageing affects semen parameters [37]. When focusing solely on men’s age, it was observed that the likelihood of pregnancy following sexual activity with men older than 34 years decreased [37]. According to a recent meta-analysis, among the features of semen analysed, sperm concentration was shown to be the least impacted by age. Still, semen volume decreased significantly with age [38]. However, Li et al. discovered that sperm concentration, total sperm count, and sperm morphology all decreased with age, although the decreases were not statistically significant [39].

The effects of father age on DNA fragmentation, sperm quality, and chromosomal abnormalities were investigated in another study [5]. Fifty fertile males (ages 25–65) had their spermatozoa compared to those of 140 infertile men (aged 24–76). Age was found to increase sperm concentration and the likelihood that sperm would be diploid [5]. However, the quality and quantity of men’s sperm appeared to decline with age. Sperm DNA fragmentation, motility, or morphology were not significantly impacted by age [5]. Separate studies evaluated the amounts of epididymal and accessory gland markers in the seminal fluid of men of varying ages and found no significant correlation with age. The drop in semen parameters was most pronounced in men older than 46 but was also noticeable in males older than 35. This resulted in a rise in the number of dead spermatozoa noticed [40]. There was an inverse relationship between male age and the features of sperm collected from men aged 30 to 40 years. Numerous other retrospective studies reveal that older men have decreased sperm volume, progressive motility, and normal morphology percentage compared to younger men [41,42,43].

### 2.3. At What Age Do the Fertility Levels Start to Drop?

Males’ sperm parameters are not noticeably altered until they are 34 years old [34]. According to research by Kidd et al., the number of sperm begins to change after a person reaches the age of 34 [34]. After the age of 40, a man’s sperm count and the percentage of his sperm that are viable begin to decline. Beyond 45 years of age, less semen is produced, and sperm motility begins to decline around the age of 43 [37]. A study based on Chinese men between the ages of 20 and 60 years found that the motility, vitality, and percentage of normal cells in their semen decreased with age.

After the age of 30, there is a sharp decline in the percentage of sperm with normal morphology [43], and progressive motility reduction and poor sperm function become more prevalent with age [44]. A possible explanation for the discrepancies between prospective and retrospective investigations is likely the participants’ age [45]. Different lengths of sexual abstinence across studies and biological variance, which makes semen parameters poor predictors of male reproductive potential, may all contribute to variations in results [46,47]. Recent research by Demirkol et al. examined the age threshold for a reduction in semen quality by analysing the changes in semen parameters across different age groups in men who presented to an infertility clinic. Among infertile males, sperm motility, morphology, and vitality declined linearly with increasing age. Sperm motility, morphology, and vitality diminish significantly beyond the age of 35. Therefore, age should be considered by infertile men who want to delay fatherhood [48].

## 3. Effects of Advanced Paternal Age on Sperm Genetic and Epigenetic Changes

### 3.1. Sperm DNA Damage

Two possible causes of sperm DNA damage are aberrant protamination and abnormal protamines compaction [49,50,51]. Infertile males exhibit an abnormal P1/P2 ratio due to protamine insufficiency [52,53], which has been linked to having 15% of histones not translated into protamines [54,55]. Oxidative stress is the cause of over 80% of the DNA fragmentation seen in situations of infection, inflammation, or different clinical diagnoses of male infertility [56,57,58]. Reactive oxygen species are more likely to develop if the antioxidants reserves are depleted, which causes oxidative stress. DNA fragmentation due to single- or double-strand breaks can be quantified with the sperm chromatin structure assay (SCSA) [59,60] or the terminal deoxynucleotidyl transferase-mediated dUTP nick-end labelling (TUNEL) test [61]. On a TUNEL assay, apoptosis and necrosis are also visible.

However, apoptosis and necrosis cannot be distinguished. Comparatively to somatic cells, the apoptosis of sperm is regulated at three different levels: the plasma membrane (via Fas receptors), the nucleus (via p53 inducing the upregulation of the Bax gene and the downregulation of Bcl-2 expression), and the cytoplasm (via the activation of Bax and release of cytochrome c and the caspase cascade in the cytosol) [62,63]. Damaged DNA and exposed phosphatidyl serine residues in the plasma membrane and mitochondria are two markers of apoptosis that may be observed in the ultrastructural examination of ejaculated sperm. Abortion-like apoptosis is evidenced by cytoplasmic fragments, incorrect chromatin packing, and/or DNA damage in immature/abnormal sperm. 

Abortive apoptosis is a process that occurs during ejaculation and allows spermatozoa that have been designated for elimination to escape death, which lowers the quality of the semen. This is primarily due to the extra cytoplasm found in sperm with morphological abnormalities [64,65,66]. Research based on the Annexin V and TUNEL assays revealed that more than 40% of the cells intended for elimination were present in the seminal ejaculate. PARP and caspase-3 activation may result in DNA damage. The diverse functions of PARP-1, such as its ability to bind nucleosomes and support the formation of compact, transcriptionally restricted chromatin structures, have been linked to both DNA damage and apoptosis. In addition, protamines cause nuclear contraction, which is related to nuclear remodelling. The process of apoptosis is set in motion when DNA damage and repair occur increasingly fast. Since PARP-1 clearance occurs before DNA damage, it serves as an early indicator of apoptosis [67]. However, apoptosis cannot account for all DNA damage in ejaculated spermatozoa [64,65,66]. Mutations and chromosomal separation can cause DNA damage and improper chromosome sorting during meiosis [68,69,70]. 

Male infertility has been linked to the intensity of DNA damage, as determined by Moskovtsev and colleagues [71]. Several studies [72,73] have shown that men’s DNA damage increases with ageing. The DNA fragmentation index (DFI) was developed by Moskovtsev et al. to quantify the DNA damage/fragmentation ratio. They discovered that men over the age of 45 had DFI values that were twice as high as those determined for men under the age of 30. Statistically, the difference is significant (15.2% vs. 32.0%). The DFI rates were from 29% to 21% to 26% in the 30–35, 35–40, and 40–45 age groups, respectively [74]. Sigh et al. found comparative results, showing that the percentage of sperm with severely damaged DNA was substantially higher in the 36–57-year-old group than in the 20–35-year-old group [72]. Another study, including 215 couples, indicated that the father’s risk of sperm DNA damage doubled between the ages of 25 and 55 years [75]. A higher DFI was seen in children whose fathers were older [72,75,76]. Males with normozoospermia had a 5% higher DFI before 40 years of age than over 40. The DFI values were 8% greater in men aged >40 than in men aged <40 in a study of individuals with oligoasthenoteratozoospermia [72]. Older fathers may have a higher risk of DNA fragmentation, according to a study by Barroso et al. [77].

In a recent meta-analysis of 26 studies including 10,220 patients, researchers found that the age of men was significantly negatively associated with DNA fragmentation [38]. Routine testing for DNA fragmentation in older men are promoted, urging patients to be aware of the risks. As expected, phosphatidylserine expression decreases with age, making it a less reliable indication of sperm quality [78]. As a signal of apoptosis, phosphatidyl serine translocation to the sperm membrane is significantly more prevalent in men over 40 years of age. A man’s age also appears to cause more sperm DNA damage.

A longer time to conception and infertility has been linked to sperm DNA damage [75,79,80]. According to studies, DNA damage is a more accurate indicator of pregnancy than typical semen characteristics [80]. Similarly, DNA damage has been linked to decreased conception rates after intrauterine insemination.

### 3.2. Telomere Analysis

Telomeres are the repetitive hexameric nucleotide sequences at the end of chromosomes (TTAGGG). Chromosome ends in eukaryotic cells are guarded by structures called telomeres. Their primary role is to ensure the genome remains unharmed [81]. Telomere shortening is a characteristic of ageing, since it happens in every cell cycle in somatic cells [82,83,84]. Previously, leukocytes were used as a surrogate for all somatic cells. Still, in recent research, Daniali et al. used four distinct types of somatic cells (leukocytes, muscle, skin and fat cells) to investigate the association between telomere length and ageing [84]. Telomere length decreased with age in the four kinds of somatic cells, not only in leukocytes [84]. Somatic cells rely on the reverse transcriptase enzyme telomerase to preserve their telomeres, which contain a guanine-rich, repetitive DNA sequence [85]. The repeats near the end of the telomeres are not entirely reproducible. Thus, some are always lost during cell division. Telomeres can be shortened, but telomerase can add TTAGGG repeats to elongate them. Telomeres shorten with ageing due to replication errors in DNA [85]. When the telomeres are too short, a cell stops dividing and either restrains the cell cycle or dies via apoptosis.

Due to its role in preserving telomere length, telomerase is most eminently expressed in highly proliferating cells, including germ cells and neoplastic cells [86,87]. Studies have found that a child’s leukocyte telomere length (LTL) increases with her/his father’s age at birth [88,89,90,91]. LTL is favourably connected with lengthened life expectancy and a decreased risk of atherosclerosis. LTL increases with paternal age, suggesting that children of older fathers may benefit from a reduced risk of atherosclerosis and a higher life expectancy [92]. Since higher LTL has been linked to a greater chance of developing breast cancer, an effect of father’s age-related LTL may also be found in daughters of elderly fathers [93].

In contrast to somatic cells, telomere length increases with age in sperm (germ cells) [94,95,96]. Despite its rarity, telomere extension may be viewed as a biological defence against ageing, even though its mechanism is poorly known. The human species’ molecular resistance to ageing may be crucial to the species’ long-term survival. This discrepancy in testicular telomerase extension has to be verified in further studies. Evidence suggests that the average telomere length may be passed down in families [90]. A recent study indicated that children’s telomere length is affected by their father’s age [97]. In recent years, data suggests paternal involvement is the critical determinant in influencing telomere length in children [98]. Over 20,000 people were included in a meta-analysis conducted by Broer et al. [8]. The degree of telomere length inheritance was assessed using data from six studies. Telomere length was inversely related to the age [8]. Male age was significantly associated with telomere length, while maternal heritability was significant for telomere length [8].

Telomeres in sperm [97] and leukocytes [91,92,97] are longer in children whose fathers are older. It is unclear how essential telomeres and telomere length are in sperm. Although there is some correlation between LTL and sperm telomere length (STL) within the same person, STL rises and LTL drops as people become older [94,95,96]. Increased quantities of reverse transcriptase, telomerase’s catalytic subunit, are most likely to be blamed for this [99,100]. When reverse transcriptase activity in germ cells is high or stem cells with shorter telomeres are eliminated by cellular attrition, longer telomeres in sperm are preferable [94,97]. It is unknown how STL affects spermatogenesis. A group of healthy persons between 18 and 19 years was analysed in research that compared telomere length to sperm count, spermatogenic activity, and parents’ age at birth [96]. The STL was strongly linked with sperm count, and men with oligozoospermia had significantly shorter STLs than normozoospermic men. Researchers also discovered that parents’ age affected their offspring’s STL [96]. According to another study that examined STL between two groups [101], men with idiopathic male infertility had shorter telomeres than controls. There are several differences between the two studies, the most important being that Thilagavathi et al. did not account for LTL, their sample size was smaller than that of Ferlin et al., their patients had a normal mean sperm count, and their patients’ ages were unknown.

The concept that telomeres play an essential role in meiosis and, by extension, safeguard genomic integrity was supported by the finding that shorter telomeres were linked with reduced spermatogenesis owing to segregation errors [102]. The testis contains primary spermatocytes in meiosis I [102]. Previous studies in our laboratory have demonstrated that primary spermatocytes very vividly have the highest levels of telomerase activity among male germ cells [103]. Short telomeres may contribute to poor spermatogenesis and male infertility, although this hypothesis has to be confirmed by other studies. Alterations in spermatogenesis may not be the root cause of shortened telomeres in ejaculated sperm. ART is affected by the finding of Thilagavathi et al. that sperm from oligozoospermic males had shorter telomeres. The pathophysiological association between STL and poor spermatogenesis and its effect on the telomere length of the progeny need to be confirmed, especially in elderly couples where the male partner is oligozoospermic (Figure 2).

### 3.3. Centrosome Aberrations

It is generally known that there is a dramatic rise in chromosomal instability due to ageing, but the mechanism responsible for this rise remains unclear. Ohshima et al. found that near-senescent human fibroblasts had a significant prevalence of mitotic abnormalities, such as mitotic slippage and incomplete mitosis [104]. There was a strong correlation between the number of centrosome abnormalities and the degree of chromosomal misalignment in metaphase cells. A centromere-specific probe analysis by fluorescent in situ hybridization demonstrated a connection between chromosomal aneusomy and centrosome over-duplication. These findings prove that increased chromosomal instability with age may be caused by aberrant duplication of the centrosomes, associated with cellular ageing [104]. The rearrangement of microtubules and centrosomes is significant during the G2/M phase. Centrosome-associated protein kinases, such as Plk [105], become less active as the cell ages [106,107]. Stem cell division slows with age, which correlates with decreased spermatogenesis [108], a phenomenon linked to centrosome misorientation (Figure 2).

### 3.4. Male Gamete Nucleus DNA Mutations

Spermatozoa undergo more cumulative cell divisions than oocytes, since they continually divide or undergo spermatogenesis during the reproductive lifetime. Several asymmetric pre-meiotic spermatogonial divisions are involved in spermatogenesis because older men’s testicles are more susceptible to the damaging effects of oxidative stress [9]. As spermatogenesis is a continuous process, spermatozoa can thus acquire de novo single nucleotide variations or mutations. De novo mutations may result from improper DNA repair and post-meiotic chromatin remodelling [109].

Furthermore, chromosomal aneuploidy was more common in the spermatozoa of older fathers [110]. According to estimates, the father’s age increases the offspring’s probability of acquiring a de novo mutation by 4% annually [111]. A sperm’s chromosomes have duplicated 150 times by the time a man is 20 years old and 840 times by the time he is 50 years old [9,112]. This higher likelihood of replication errors in the germ line increases the de novo mutation rate in the spermatozoa [111]. The problem is worsened by damage to age-sensitive systems, such as DNA replication and repair. Kong et al. found that the prevalence of de novo mutations rises with age [111]. The average annual de novo mutation rate increase is of about two base pairs [111]. Kong et al. found that the father’s age is a significant factor in the heritability of mutations in the offspring [111]. This increases the likelihood of a genetic disease being transmitted from an old father to his offspring [113]. Mutations in FGFR (fibroblast growth factor receptor) are the root cause of paternal age effect (PAE) diseases [114,115,116]. A study conducted by Wyrobek et al. discovered that sperm from men between the ages of 22 and 80 contained an FGFR3 mutation, which was linked to achondroplasia [117]. A significant contributor to human mutations is the increased age of the paternal population [9]. Although this phenomenon promotes species diversity, it might, regrettably, also lead to a rise in the prevalence of uncommon diseases in human beings. Microdeletions of the long arm of the Y chromosome affect 10% of azoospermic or severely oligozoospermic males, while 5% of all infertile males have chromosomal abnormalities such as Klinefelter syndrome, indicated as 47,XXY [118]. Studies suggest that a de novo genetic mutation can occur in sperm during the post-meiotic stages [109]. Numerous other mechanisms have been proposed to account for these de novo mutations. One is a base substitution due to polymerase failure to integrate nucleotides [119], and another is an insertion and deletion that can lead to fast cell division and subsequent de novo mutations [120].

A de novo chromatin translocation was found in 10 sperm donors, and its occurrence and expansion were not dependent on the donors’ age [121]. This finding is significant because it shows a replication-independent method for producing translocations. In addition to its other functions, nuclear receptor NRA51 is well recognised as the master regulator of steroidogenesis. In 4% of males with severe spermatogenic failure, for which other reasons have been ruled out, mutations in NRE5-1 were discovered [122]. These conditions include primary ovarian insufficiency and anomalies in sex development in 46,XY individuals. Testicular dysgenesis is a possible cause of infertility in males who report low or fluctuating levels of testosterone and gonadotropins; a complete clinical evaluation is required owing to the existence of the male factor. A de novo point mutation in the USP9Y gene on the Y chromosome causes non-obstructive azoospermia, as was recently discovered in a male [123]. Like others before it, this study found that deleting a single gene was responsible for sterility (Figure 2).

### 3.5. Chromosomal Abnormalities

A cell with an abnormal number of chromosomes is said to be chromosomally aneuploid. When a sperm cell performs meiosis and the chromosomes are unequally split into the daughter cells due to disjunction, chromosomal aneuploidy results. Chromosomal aneuploidy is the most common cause of miscarriage, since most aneuploid embryos die during pregnancy [124]. However, the 1% of live births from aneuploid pregnancies [124] may be associated with multiple congenital disabilities and/or mental retardation [125]. 

About 10% of aneuploid sperm cells in a healthy male population have an extra copy of chromosomes 21 and 22 [126]. However, that number rises as fathers are older [127]. When comparing males of different ages, there is a striking rise in the prevalence of sex chromosome and chromosome 18 disomy among men older than 50 years [127]. Compared to fathers aged 25–29 years, McIntosh et al. found a twofold increase in this risk among those aged 50 and higher [128]. 

### 3.6. Instability in Molecular Structure and Aeging-Related Reduction in Gene Function

Ageing is a complicated process that involves many factors and leads to a gradual loss of cellular function and a higher risk of disease [129]. Several processes are impacted by ageing, resulting, for example, in DNA damage [130] and telomere shortening [131,132] that lead to cellular senescence or apoptosis [132]. In this setting, a father’s advanced age is associated with an increased likelihood of autism and schizophrenia in his children and with male infertility. In cellular senescence, the DNA damage response was reported to be induced by the telomerase malfunction [133].

Changes in the microRNA (miRNA) pattern and variations in gene expression are two outcomes of ageing resulting from genomic instability at the cellular level [132,134,135]. To control gene expression after transcription, miRNAs drive messenger RNAs (mRNAs) toward cleavage or translational suppression. These miRNAs are expressed differently as people age or experience stressful life events such as a vasectomy [136]. They have been detected in male reproductive fluids such as the seminal plasma. Because of this, further study is needed to identify the genes involved in normal sperm and abnormal sperm development in adults and the elderly. Spermatozoa complete their growth and maturation in the epididymis, where they are transported and stored and mature, by means of a network of coiled tubes [137,138]. Zhang et al. examined the expression of miRNAs in the epididymis of newborns, adults (over the age of 25), and older men (aged 75 years). The study found that 251 miRNAs (representing 63% of the known miRNAs) were expressed in the neonatal epididymis, whereas only 31% were expressed in the elderly epididymis [139]. Still unknown is how miRNA expression changes affect sperm quality and DNA stability. 

### 3.7. Epigenetics Alterations during Male Ageing

Epigenetics refers to a heritable, persistent change in chromatin structure at the level of histone tails rather than the DNA sequence, which results in differential gene expression [140]. Contrary to DNA mutations, epigenetic patterns can be altered or silenced by environmental and endogenous factors, such as phenotypic variety, age, nutrition, and drug/toxin exposure. Therefore, the male gamete undergoes epigenetic reprogramming at several stages during spermatogenesis and spermiogenesis, with each stage being affected by a unique combination of environmental variables. Epigenetic processes can interfere with fertilisation, implantation, and embryonic development [141]. Men with low sperm counts for no apparent reason may have an underlying problem with methylation of the paternally imprinted H19-DMR (differentially methylated region) gene [142]. Nearly 20% of males with moderate to severe oligozoospermia have epigenetic abnormalities [143,144,145]. Evidence for extensive DNA hyper-methylation was found in low-quality sperm after a whole-genome analysis [146]. This suggests that DNA methylation was not adequately erased during the development of germ cells. Changes or extensions of the imprinted congenital phenotype have been connected to epigenetic modifications in sperm utilised for ART selection. Methylation at two imprinted loci, H19-DMR and PEG 1/MEST-DMR, was examined in men whose symptoms ranged from severe oligozoospermia to normospermia. Montjean and colleagues used the methylation profile of these two loci to measure sperm DNA methylation state [147]. Genetic variants were found in 20% of H19-DMR and 3% of PEG 1/MEST-DMR spermatozoa of oligozoospermic males, but no link was found between these variations and the ART outcome.

The epigenetic traits of a man’s children and grandchildren are shaped more by food and exposure to toxicants than by paternal age alone [10]. Benchaib et al. found no correlation between DNA methylation and paternal age [148,149]. Some epigenetic factors appeared heritable and stable in this study. Two of the most typical methods for silencing genes are DNA methylation and repressive histone modification. The effects of DNA methylation on mammalian development, including X-inactivation [150], genomic imprinting, and early embryonic development after zygote formation [151], have recently been uncovered.

To further demonstrate the significance of DNA methylation in embryo development, Benchaib et al. undertook prospective research to evaluate the impact of global sperm DNA methylation on in vitro fertilisation (IVF) success rates [149]. The researchers discovered that the success rate of pregnancies was dramatically improved by using sperm with a global methylation level (GML) over an arbitrary threshold number (555 AU). However, fertility rates and embryo quality were not different, according to other studies [148,149]. These researchers hypothesize that spermatogenesis issues and male infertility may result from epigenetically altered germ lines. Ace-1(Ace-variant1), prm1 (Protamine 1), prm2 (Protamine 2), and smcp are essential sperm genes that bind to chromatin (sperm mitochondrial-associated cysteine-rich protein). Ace-1, Prm1, Prm2, and Smcp are epigenetic genes whose expression levels decreased with increasing paternal age in a recent longitudinal study in mice. During spermiogenesis, these proteins are used instead of histones [151]. Sperm quality and IVF success are negatively impacted by low levels of Prm1, Prm2 or both [152].

Gene silencing is also triggered by an increase in the methylated forms of cytosine (5-mc and 5-hmc) with the increase of paternal donor age (1.76% per year) [153]. Angelman syndrome, a neurogenetic condition, has been associated with delayed intellectual development [154,155]. In contrast, Bechwith–Wiedmann syndrome has been linked to unusually early puberty and an increased risk of cancer in children. According to research by Gosden et al., newborns conceived by ART had a much higher frequency of uncommon disorders such as Angelman syndrome and Beckwith–Wiedemann syndrome [156]. This finding may be attributable to older couples turning to ART to start their families (Figure 1).

### 3.8. Wide-Scale Genome Investigations

Genome-wide association studies (GWAS) focus on many genetic changes across many genomes to find those statistically associated with a specific trait or disease. It has been reported that over a thousand genes have been associated with male infertility [157]. These genes’ transcriptome, genomic, and epigenomic activity must be better understood. However, several single-nucleotide polymorphisms (SNPs) and transcripts vary during male reproductive age. Understanding the molecular mechanism and signalling pathways involved in male reproductive function, generally and specifically, as it relates to ageing requires gene discovery based on hybridisation/microarrays technologies followed by specific target identification utilising throughput sequencing [158]. Through the examination (sequencing) of the DNA, RNA, miRNA, SNPs, copy number variations (CNVs), insertions/deletions and other genomic parameters associated with male infertility and aging, the GWAS method has the potential to shed light on a wide variety of genetic diseases [113,159,160]. However, due to sample size limitations and the heterogeneity of the ageing effects, conducting research of this kind is difficult [129]. Functional validation necessitates a thorough comprehension of the molecular mechanisms underlying male infertility at a certain age and across time [157].

## 4. Advanced Paternal Age Effect on Reproductive and Fertility Outcome

### 4.1. Decreased Pregnancy Rate in Assisted Reproductive Technology

According to research, the odds of successful intrauterine insemination (IUI) or a live birth decrease as men become older [161,162,163]. Mathieu et al. found that the clinical pregnancy rate was lower with fathers over 35 years [161]. A significant fall in artificial conception rate was found by Belloc et al., who showed that the pregnancy rate dropped from 12.3% per cycle with males aged 30–44 years to 9.3% with men aged 45–54 [162]. Demir et al. also found comparable results [163]. A decrease in the incidence of artificial pregnancies was reported in a prospective study by Klonoff-Cohen et al., showing that as paternal age increased, the live birth rate dropped [164]. Nij and his colleagues investigated a correlation between semen qualities and the age of men in a prospective study of 278 males and women who had undergone intracytoplasmic sperm injection (ICSI) or IVF. The relationship between a man’s age and fertility was shown to be non-existent [165]. The inability of assisted reproductive technology (ART) to result in pregnancy or avoid miscarriage has been connected to rising DNA fragmentation rates and declining sperm motility [166]. This is supported by the finding that there is an association between older paternal ages (particularly, for fathers over 40 years old) and decreased DNA integrity [72], which has a detrimental effect on the success rate of IVF/ICSI and, by extension, ART outcomes [159,167]. Sperm DNA integrity is essential for successful IVF and healthy embryo development. Recent research indicates that paternal ageing hinders the early embryonic development because of its effect on sperm DNA integrity. Morris et al. discovered that sperm DNA damage significantly correlates with reduced post-fertilization embryo cleavage for males aged 29 to 44 years [168]. A further study with 132 ICSI patients demonstrated that post-implantation sperm DNA fragmentation was significantly altered throughout embryonic development [169]. This was true even though all the fathers were under 40 years of age. A cross-sectional study of 215 infertile men who had undergone ART found that increased DNA damage in sperm significantly negatively influenced the early embryonic development and considerably reduced the eventual embryo implantation [170]. A study by Frattarelli et al. on 1023 infertile couples found that older men’s sperm resulted in a normal early clearage of the embryo but a lower blastocyst formation rate [171].

Two extensive studies found a correlation between paternal age and the risk of pregnancy loss following a successful IUI pregnancy, suggesting that paternal age may substantially influence embryo development [172,173]. The success of ICSI or IVF in conceiving a child is not correlated with paternal age [172,173,174,175]. However, a meta-analysis of seven IVF and IVF/ICSI studies found no association between paternal age and subsequent pregnancy loss [175]. This might be because men who gave birth without medical intervention or after IUI were likelier to have normal semen parameters. In contrast, men who had children following IVF/ICSI were chosen from a pool with more diverse demographic data, which lessened the impact of age. The increased DNA fragmentation in sperm that happens with paternal age significantly influences the outcomes of IVF/ICSI, the implications of ART, and early embryo development. The American Society for Reproductive Medicine (ASRM) guidelines do not advise to routinely measure sperm DNA integrity in clinical practice, despite increasing evidence correlating high levels of sperm DNA fragmentation to enhanced male fertility [176]. Researchers emphasize that more extensive, randomized studies are required to validate the sperm DNA integrity test.

### 4.2. Pregnancy Loss

Miscarriage is spontaneous pregnancy loss before the 20th week [126]. The older the father, the higher the chance the pregnancy will result in a spontaneous abortion [177]. According to a retrospective analysis by de la Rochebrochard and coworkers, the odds ratio of having a miscarriage increased from 1.06 to 1.31 to 1.80 when paternal age increased from 30–34 years, to 35–39 years, to 40–64 years, respectively, while maternal age remained unchanged at 20–29 years [177]. After accounting for maternal age, other studies found similar increases in the risk of miscarriage for fathers between the ages of 30 and 34 and 35 and 39 years and for those 40 years old and older (OR = 1.22, 1.50, and 1.30, respectively) [178]. According to the findings of Slama et al., participants in the age group older than 35 years had a 1.27-fold greater risk compared to those in the younger age group who were younger than 35 years [179]. A recent French study by Belloc et al. found that the miscarriage rate increased from 13.7% to 32.4% for fathers aged 45 years and older compared to fathers aged 30 years and younger [162]. Nevertheless, the outcomes of several other studies varied [164,175,180,181].

### 4.3. Premature Birth

If a baby is born before 37 weeks of gestation, it is considered premature [182]. Premature birth is the leading cause of infant mortality, responsible for more than a million stillbirths annually [183]. According to research by Zhu et al., a greater risk of preterm birth was shown to occur when the father was older [184]. Preterm birth was more likely with fathers aged 25–29, 35–39, 40–44, 45–49, and >50 years than with fathers aged 20–24 years [184]. Astolfi et al. found that the probability of preterm birth increases with the father’s age [185]. After accounting for the mother’s age, the researchers discovered that the odds increased by a factor of 1.91 when the mother was 20–24 years old and by 1.71 when she was 25–29 years old [185]. However, several earlier studies found no indication that paternal age was associated with a greater risk of premature birth [186,187,188,189].

### 4.4. Low Birth Weight

Low birth weight in the United States is a common cause of premature death. Chronic lung disease, attention deficit hyperactivity disorder, blindness, epilepsy, and cerebral palsy are only some long-term health problems [190]. According to research by Alio et al., fathers older than 45 had a 19% higher chance of having a baby with a low birth weight and a 13% higher risk of having a baby prematurely (born between 33 and 37 weeks of gestation) [191]. In another study, Reichman et al. [9] conducted a cohort analysis and found that compared to fathers aged 20–34 years, those aged 35 years and older had a 1.9-fold increased risk of having children born with low birth weight.

### 4.5. Stillbirth

Alio et al. found that the risk of stillbirth (in utero foetal death at 28 weeks) was 48% greater for the paternal age group >45 years than for the paternal age group of 25–29 years [191,192]. Nybo et al. found that when maternal age was adjusted, the probability of late foetal mortality (after 20 weeks) was raised by an odds ratio of 1.40 in pregnancies where the father was 45 to 49 years old [193]. When fathered by men over the age of 50, there was an elevated risk of both early foetal mortality (before 20 weeks of gestation) and late foetal death (beyond 20 weeks of pregnancy) (hazard ratio, 1.38 and 3.94, respectively) [193]. Alio et al., when comparing fathers aged 25–29 and 45+ years, found that the risk of having a stillborn child increased by 22% [191]. According to research by Astolfi et al., the risk of having a stillborn child increases with paternal age [194].

## 5. Advanced Paternal Age Effect on the Offspring

Strong evidence links older fathers to an increased risk of passing on certain genetic diseases to their offspring. It has been hypothesised that paternal age impacts diseases resulting from an increase in the de novo mutation rate. Errors in DNA replication lead to single-gene abnormalities in sperm. These are called sentinel phenotypes because they are diseases with poor fitness and high frequency that result from highly penetrant mutations. One possible explanation for accumulating new single-gene mutations linked to age is the continuous cell division taking place during spermatogenesis. Errors in DNA transcription, which cause single-gene mutations, may be more likely to occur in the spermatogonia of older men due to their having experienced several cell divisions [195]. Kong et al. performed deep sequencing analysis research and found that germline single base-pair alterations increased at a rate of two base pairs per year as paternal age increased [111].

The British Andrology Society and the ASRM have established a maximum age of 45 years for sperm donors due to the seriousness of paternal age diseases in males [196,197]. Here, we will discuss some genetic problems linked to advanced paternal age.

### 5.1. Schizophrenia

Multiple studies have shown an association between paternal age and schizophrenia [198,199,200,201]. Schizophrenia is characterised by psychotic symptoms that may be either permanent or intermittent [202]. Schizophrenia is a complex disorder with several possible causes and a significant hereditary component [203,204]. There is such a significant hereditary component that a rising paternal age is implicated in one-fourth of all occurrences of schizophrenia [205]. It was shown in a cohort study involving 754,330 Swedish individuals that the probability of having a child with schizophrenia increased by 1.47 times for every decade of the father’s age at conception [206].

Miller et al. performed a meta-analysis of 12 separate cohort and case–control studies. They found that compared to the paternal reference age of 25–29 years, children born to fathers older than 30 had increased odds of schizophrenia [200]. Tsuchiya et al. researched Japan to see whether there was a close connection between advanced paternal age and schizophrenia in a socially and culturally distinct population [207].

The researchers Frans et al. wanted to know whether grandparents’ age affected the likelihood that their grandchild would develop schizophrenia. They surveyed 120,758 people from a nationwide register-based cohort study. It was shown that having a very elderly grandmother raised the chance of schizophrenia in a grandchild, whereas having a very old grandfather did not [201]. It has been suggested that the accumulation of de novo mutations in sperm is the primary cause of the correlation between paternal age and schizophrenia [201,203,204,206].

Studies have pointed to de novo mutations as a possible cause of schizophrenia, although other processes may also play a role. For instance, when considering paternity for the first time, there was no longer a correlation between paternal age and the risk of schizophrenia [208]. The schizophrenia risk may also be increased by the dysregulation of epigenetics related to paternal age, namely, of DNA methylation, histone alterations, and chromatin remodelling. Genomic imprinting, sometimes called parental imprinting, occurs when a gene shows differential expression depending on which parent supplied the related DNA [209]. Modifications to parental imprinting and other epigenetic pathways may affect the offspring [210].

### 5.2. Bipolar Disorder

Bipolar disorder is a mental disease marked by depression and manic episodes [211]. Multiple studies have shown a strong correlation between father’s age and his offspring’s risk for bipolar disorder [212,213]. For example, according to Frans et al., who analysed data from a population-based registry of 7328 and 100 people, the risk of bipolar illness in children rose with the age of the father [212].

The likelihood of a bipolar disorder diagnosis for children with fathers aged 55 years and older was 1.34 times greater than for those with fathers aged 20–24 years (the control group) [212]. After accounting for maternal age, Menezes et al. found that the risk factor for bipolar disorder in the offspring increased by 1.20 times for every decade of paternal age [213].

A recent population-based cohort research comprising 2,615,081 people from Sweden observed that a higher paternal age was related to an increased risk of bipolar disorder. Compared to children born to parents aged 20–24 years, those born to parents aged 45 and older had a high hazard ratio (or relative risk ratio) of 24.7 [13]. However, Buizer-Voskamp et al. found no correlation between fathers’ advanced age and a higher incidence of bipolar disorder in the offspring [199].

De novo mutations, the consequence of DNA copy defects, may be the origin of bipolar disorder, just as they are for schizophrenia and other mental diseases linked to older fathers. Conditions influenced by the paternal age may also have an epigenetic component [214].

According to research by Kaminsky et al., patients with bipolar disorder had higher levels of DNA methylation in the human leukocyte antigen complex group 9 gene (HCG9) compared to the control group [215]. Since DNA methylation increases with old age, this may shed light on the likely mechanism by which the offspring of older fathers are at a greater risk of developing bipolar disorder.

### 5.3. Autism

The term “autism spectrum disorder” describes conditions marked by communication and socialisation issues and a propensity toward repetitive behaviours [216]. Strong evidence links older fathers to a higher autism incidence rate in the offspring [13,199,217,218,219].

It was determined in a recent registry research by Buizer-Voskamp and colleagues that older men (>45 years) had 3.3 times the chance of having a child with autism when compared to younger fathers (20 years) [199]. Reichenberg et al. found that the risk of autism in children born to parents aged 50 years or older was 5.75 times that in children born to parents aged 30 [217]. A meta-analysis by Hultman et al. discovered that as fathers became older, their children were more likely to develop autism [218]. Mutations in transcription factors are thought to impact gene expression significantly and may be one cause of autism [216].

The increased risk of autism is seen throughout generations, which may explain the substantial heritability of epigenetic variables. In addition, according to some research, the likelihood of neurodevelopmental problems such as autism may also be enhanced by de novo mutations in male germ as people age [220,221].

### 5.4. Childhood Cancer

A substantial linear dose–response connection was found between paternal age and the incidence of hematologic malignancies in children, with children of males older than 35 having a 63% greater risk than those whose fathers were younger than 25 years of age [222]. Among almost two million children born in Denmark between 1978 and 2010, researchers found that the risk of childhood acute lymphoblastic leukaemia increased by 13% for every 5-year increment in the father’s age [223]. Cancers of the breast and central nervous system are two examples of child cancers linked to father’s advanced age [224,225,226]. Lengthening telomeres is a potential mechanism through which paternal ageing contributes to an increased risk of cancer [92].

The shortening of telomeres is linked to many illnesses and is considered a lifespan barrier. For example, children of older fathers (ages 62–64 years) showed leukocyte telomeres extended by 0.5–2 times for every year of paternal age increase. While this may positively affect health and lifespan, a greater cancer risk has been observed [92,227].

### 5.5. Other Disorders

The impact of father’s age on neurocognitive problems was examined. Other disorders, including autosomal disorders such as achondroplasia and Apert syndrome and congenital abnormalities such as Klinefelter syndrome, have also been linked to older fathers.

## 6. Conclusions

Male infertility is associated with the molecular aging process, which has been shown to affect sperm quality and alter the profile of reproductive hormones. Numerous studies have shown that paternal age affects various biological processes, such as oxidative stress, DNA mutations, chromosomal abnormalities, Y chromosome microdeletions, telomere elongation, centromere aberrations, epigenetic patterns, and miRNA expression abnormalities. Due to these factors, male fertility decreases with age. These changes may be associated with adverse pregnancy outcomes such as stillbirths, spontaneous abortions, preterm births, and low birth weight. Children with older fathers are at a higher risk for genetic abnormalities, paediatric malignancies, and neuropsychiatric problems.

## 7. Future Directions

Despite extensive research on the adverse effects of paternal age, our understanding of the molecular pathways that produce these effects still needs to be improved. It is suggested that more studies be undertaken to understand the underlying causes better. The development of cutting-edge diagnostic and therapeutic tools for male infertility will be facilitated by further investigating the molecular mechanisms underlying reproductive ageing, which can be reduced by delineating the relationship between ageing and male infertility using state-of-the-art technologies such as next-generation sequencing. With the help of the discovered genes for ageing and longevity, the impact of old age on human reproductive ability may be evaluated. It is possible that a reasonable “age threshold” might be defined as well, beyond which a prospective father would be consulted to addressed to specialized clinics for management and risk assessment at an advanced age.

## Figures and Tables

**Figure 1 genes-14-00486-f001:**
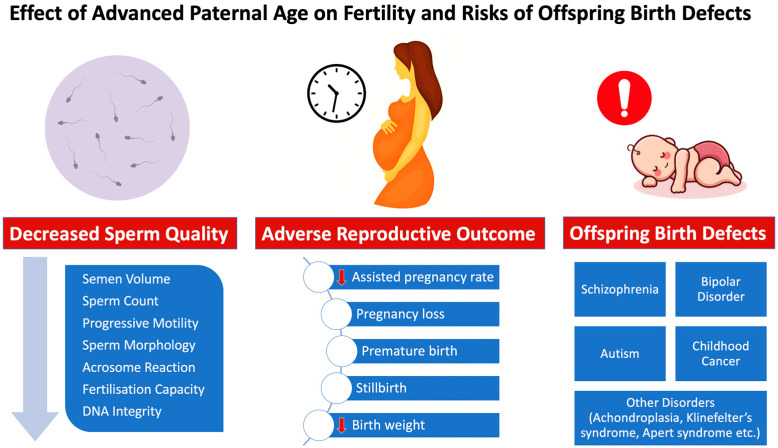
Effects of increased paternal age on male fertility status, reproductive/fertility outcome and risk of offspring birth defects.

**Figure 2 genes-14-00486-f002:**
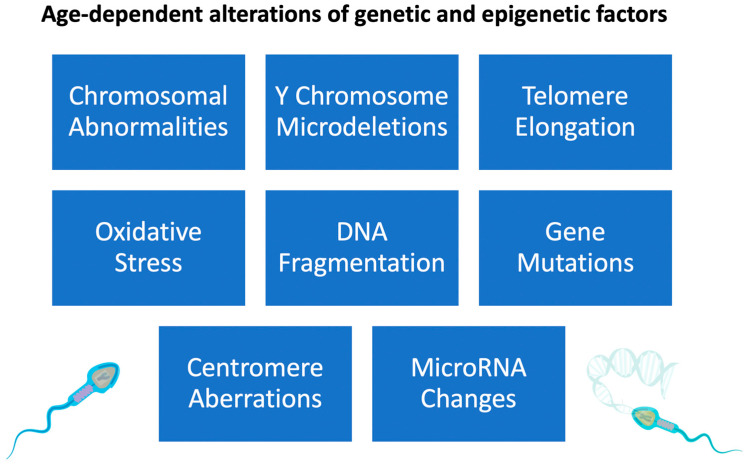
Effects of increased paternal age on sperm genetic and epigenetic changes.

## Data Availability

No new data were created or analysed in this review study. Data haring id not applicable to this article.

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
