# Peer review of "Impact of Advanced Paternal Age on Fertility and Risks of Genetic Disorders in Offspring"

_genes, 2023, doi:10.3390/genes14020486_

Round 1

Reviewer 1 Report

This review summarized the effects of advanced paternal age on sperm and testis function, changes of sperm genetics and epigenetics, outcomes of fertility and offspring health. Although the progress in this field is emerging slowly and more solid evidence are needed in the future studies, the review has important implications for understanding the reproductive risks of advanced age. I have several minor comments or suggestions to improve the quality of the manuscript.

1. The content of the article is too tedious. Some irrelevant contents should be condensed or deleted, such as in the section “3.4 centrosome aberrations”.

2. Section 6. Conclusions are inadequate. It did not contain some important points authors summarized.

3. Line 299, dose the “STL” refers to “sperm transit time” in line 307? It did not be defined in advance. Also, what is the FSH in line 83? Similar mistakes should be examined and corrected in detail.

4. Figures should be carefully re-crafted. The lines, fonts, and content are too rough. In addition, the legends of the Figures 1 and 2 are too short and should be elaborated.

5. The section “2. Advanced Paternal Age Effects on Sperm Quality and Testicular Function” should be changed to “2. Advanced Paternal Age Effects on Testicular Function and Sperm Quality”. This will better correspond to the sequence of the following paragraphs.

6. I suggest the section “3.2. Sperm DNA Damage and the Outcomes of Assisted Reproductive Technology” should be a part of the section “3.1. Sperm DNA Damage”. This is more in line with the overall idea of the manuscript. Some of the relevant findings about Assisted Reproductive Technology from this section can be removed and included in the section “4.1. Decreased Pregnancy Rate in Assisted Reproductive Technology”.

7. The section “4.3. Pre-eclampsia” doesn't fit in well with the whole and should be deleted. Pre-eclampsia is a type of hypertensive disorder complicating pregnancy. It has more to do with maternal hypertension than reproduction.

8. While the literature was reasonably presented, some important articles, especially several very recent publications, are missing, for example:

1) Testicular aging, male fertility and beyond. Front Endocrinol (Lausanne). 2022 Oct 13; 13:1012119

1) Single-cell analysis of human testis aging and correlation with elevated body mass index. Dev Cell. 2022 May 9;57(9):1160-1176.e5.

These references should be added and further discussed.

Author Response

Response to Reviewer 1 Comments

  1. The content of the article is too tedious. Some irrelevant contents should be condensed or deleted, such as in the section “3.4 centrosome aberrations”.

Response: Thank you for your valuable comments and suggestions. We have shortened the section '3.4 Centrosome aberrations" according to your recommendation.

  1. Section 6. Conclusions are inadequate. It did not contain some important points authors summarized.

Response: Thank you very much for your insightful remarks and recommendations. We have carefully rewritten the conclusion to include all relevant details.

  1. Line 299, dose the “STL” refers to “sperm transit time” in line 307? It did not be defined in advance. Also, what is the FSH in line 83? Similar mistakes should be examined and corrected in detail.

Response: We thank you for your valuable comments and apologize for our mistakes. We have corrected all similar mistakes.

  1. Figures should be carefully re-crafted. The lines, fonts, and content are too rough. In addition, the legends of the Figures 1 and 2 are too short and should be elaborated.

Response: Thank you so much for your insightful remarks and comments. To help the reader better understand the meaning of the manuscript, we have created completely new illustrations.

  1. The section “2. Advanced Paternal Age Effects on Sperm Quality and Testicular Function” should be changed to “2. Advanced Paternal Age Effects on Testicular Function and Sperm Quality”. This will better correspond to the sequence of the following paragraphs.

Response: Thank you for your valuable comments and suggestions. We have made the changes you suggested to improve the article for the benefit of the reader.

  1. I suggest the section “3.2. Sperm DNA Damage and the Outcomes of Assisted Reproductive Technology” should be a part of the section “3.1. Sperm DNA Damage”. This is more in line with the overall idea of the manuscript. Some of the relevant findings about Assisted Reproductive Technology from this section can be removed and included in the section “4.1. Decreased Pregnancy Rate in Assisted Reproductive Technology”.

Response: We appreciate your feedback and ideas. After reviewing your feedback, we have implemented your suggestions to improve the content.

  1. The section “4.3. Pre-eclampsia” doesn't fit in well with the whole and should be deleted. Pre-eclampsia is a type of hypertensive disorder complicating pregnancy. It has more to do with maternal hypertension than reproduction.
    Response: In response to your comment, we have deleted this section.
    8. While the literature was reasonably presented, some important articles, especially several very recent publications, are missing, for example: 1) Testicular aging, male fertility and beyond. Front Endocrinol (Lausanne). 2022 Oct 13; 13:1012119; 2) Single-cell analysis of human testis aging and correlation with elevated body mass index. Dev Cell. 2022 May 9;57(9):1160-1176.e5. These references should be added and further discussed.
    Response: Thank you very much for your valuable comments and suggestions. Your suggestions for two articles were included into the main body of the content. Many thanks to the reviewers for their approval of our manuscript.

Reviewer 2 Report

The review article is a significant contribution to the field of “Paternal ageing effects on sperm and next-generation offspring". However, some paragraphs could be rewritten and structured more precisely. Please find comments below:

1. What is the main question addressed by the research?
The review article mainly discusses the effects of advanced paternal age on sperm, reproductive and fertility outcome, and also the effects on the next-generation offspring.

2. Do you consider the topic original or relevant in the field? Does it address a specific gap in the field?
Yes, the reported review is original and relevant to the area of parental ageing and its consequences to germline and next-generation children. Since it is a review article and not an original research work, it does not address any specific gap in the field.

3. What does it add to the subject area compared with other published material?
This review summarizes the work done by other researchers in the field of ageing. It specifically elaborates on male ageing effects on his germline, male hormones, and impact on reproduction. Work on both genetic and epigenetic changes during male ageing has been discussed. Research articles regarding telomere analysis, centrosome aberrations, sperm nucleus DNA mutations, and chromosomal abnormalities have been cited. Furthermore, the impact of paternal ageing on assisted reproductive technology and the health of the offspring are illustrated.

4. What specific improvements should the authors consider regarding the methodology? What further controls should be considered?
I would like to suggest that the author should consider rephrasing some paragraphs so that it becomes easy for the reader to follow. Since this is not an original research work, there are no improvements to be considered in methodology or controls.

5. Are the conclusions consistent with the evidence and arguments presented and do they address the main question posed?
Yes, the conclusions are consistent with the evidence and arguments presented and they do address the main question posted.

6. Are the references appropriate?
Yes, the given references are appropriate.

7. Please include any additional comments on the tables and figures.
Figure 3 is a good explanation of advanced paternal age effects. However, figure 1 and 2 can be improved. In fact, figure 2 does not really explain what the author would like to convey. Figure 2 should be removed or modified to a better version. The author can consider elaborating well on the figures (especially figures 1 and 2).

The review article is good to be published after minor modifications in the figures and also after rephrasing some paragraphs. 

Author Response

Response to Reviewer 2 Comments

1.What is the main question addressed by the research? 
The review article mainly discusses the effects of advanced paternal age on sperm, reproductive and fertility outcome, and also the effects on the next-generation offspring. 
2. Do you consider the topic original or relevant in the field? Does it address a specific gap in the field? 
Yes, the reported review is original and relevant to the area of parental ageing and its consequences to germline and next-generation children. Since it is a review article and not an original research work, it does not address any specific gap in the field. 
3. What does it add to the subject area compared with other published material? 
This review summarizes the work done by other researchers in the field of ageing. It specifically elaborates on male ageing effects on his germline, male hormones, and impact on reproduction. Work on both genetic and epigenetic changes during male ageing has been discussed. Research articles regarding telomere analysis, centrosome aberrations, sperm nucleus DNA mutations, and chromosomal abnormalities have been cited. Furthermore, the impact of paternal ageing on assisted reproductive technology and the health of the offspring are illustrated. 
4. What specific improvements should the authors consider regarding the methodology? What further controls should be considered? 
I would like to suggest that the author should consider rephrasing some paragraphs so that it becomes easy for the reader to follow. Since this is not an original research work, there are no improvements to be considered in methodology or controls. 
5. Are the conclusions consistent with the evidence and arguments presented and do they address the main question posed? 
Yes, the conclusions are consistent with the evidence and arguments presented and they do address the main question posted. 
6. Are the references appropriate? 
Yes, the given references are appropriate. 
7. Please include any additional comments on the tables and figures. 
Figure 3 is a good explanation of advanced paternal age effects. However, figure 1 and 2 can be improved. In fact, figure 2 does not really explain what the author would like to convey. Figure 2 should be removed or modified to a better version. The author can consider elaborating well on the figures (especially figures 1 and 2). 
The review article is good to be published after minor modifications in the figures and also after rephrasing some paragraphs. 

Response: Thank you for your valuable comments and suggestions. In order to help the reader better understand the meaning of the manuscript, we have created completely new illustrations and shortened some paragraphs. Many thanks to the reviewers for their approval of our manuscript.
